# eDNA captures depth partitioning in a kelp forest ecosystem

**Keira Monuki** ¤*, **Paul H. Barber, Zachary Gold**

Ecology and Evolutionary Biology, University of California, Los Angeles, California, United States of America

¤ Current address: Bodega Marine Laboratory, University of California Davis, Bodega Bay, California, United States of America
* ksmonuki@ucdavis.edu

**Data Availability Statement:** All data is available through Dryad (https://doi.org/10.5068/D18H47). Code used for analyses is available on GitHub at https://github.com/ksmonuki/eDNA-analysis. A Zenodo link for the code is available at https://doi.org/10.5281/zenodo.5574141. The raw sequences are available on NCBI at http://www.ncbi.nlm.nih.

## Abstract

Environmental DNA (eDNA) metabarcoding is an increasingly important tool for surveying biodiversity in marine ecosystems. However, the scale of temporal and spatial variability in eDNA signatures, and how this variation may impact eDNA-based marine biodiversity assessments, remains uncertain. To address this question, we systematically examined variation in vertebrate eDNA signatures across depth (0 m to 10 m) and horizontal space (nearshore kelp forest and surf zone) over three successive days in Southern California. Across a broad range of teleost fish and elasmobranchs, results showed significant variation in species richness and community assemblages between surface and depth, reflecting microhabitat depth preferences of common Southern California nearshore rocky reef taxa. Community assemblages between nearshore and surf zone sampling stations at the same depth also differed significantly, consistent with known habitat preferences. Additionally, assemblages also varied across three sampling days, but 69% of habitat preferences remained consistent. Results highlight the sensitivity of eDNA in capturing fine-scale vertical, horizontal, and temporal variation in marine vertebrate communities, demonstrating the ability of eDNA to capture a highly localized snapshot of marine biodiversity in dynamic coastal environments.

## Introduction

Environmental DNA, or eDNA, is a metabarcoding approach for detecting and cataloging local community biodiversity, and is increasingly used in marine habitats [1,2]. eDNA approaches rely on the capture of freely dissociated cells/DNA originating from organisms in the environment, which can then be isolated and sequenced to reconstruct the community of organisms present [1,3]. Previous applications of eDNA metabarcoding in marine ecosystems demonstrate its effectiveness for measuring biodiversity and capturing important ecological information about taxa ranging from microbes to mammals [1,2], making it particularly useful for ecosystem monitoring.

The rapid rise of eDNA metabarcoding in marine ecosystems is driven by several key advantages of this technique. First, eDNA is cost-effective and has the potential for automation, allowing for increased sampling effort compared to traditional surveys [4,5]. Second,

gov/bioproject/771917 with accession number
PRJNA771917.

**Funding:** Funding was provided by the UCLA
Ecology and Evolutionary Biology Whitcome
Undergraduate Summer Research Fellowship and
the UCLA Honors Programs Irving and Jean Stone
Research Award (http://www.honors.ucla.edu/
honors-summer-research-stipends/). K.M. was
supported by the UCLA Undergraduate Research
Scholars Program (http://sciences.ugresearch.ucla.
edu/programs-and-scholarships/ursp/), Z.G. was
supported by the US-NSF Graduate Research
Fellowship, grant number DEG No. 1650604, and
an HHMI professor award to P.H.B. The funders
had no role in study design, data collection and
analysis, decision to publish, or preparation of the
manuscript.

**Competing interests:** The authors have declared
that no competing interests exist.

DNA barcoding approaches do not require extensive taxonomic expertise, allowing for the
identification of a broad range of marine taxa from a single sample. Third, eDNA sampling
only requires the collection of sea water, eliminating the potential risks associated with
repeated dives needed for SCUBA-based visual surveys [4,6].

Despite these advantages, because eDNA is a relatively new technique, we know less about
methodological biases compared to well-established visual survey methods. In particular, key
questions remain about the temporal and spatial variability of eDNA signatures in marine
environments, information that is critical for the effective design of eDNA biomonitoring
efforts. Previous studies in dynamic freshwater ecosystems show daily variation in eDNA sig-
natures [7], yet little is known about temporal variation in marine ecosystems. Degradation of
eDNA signatures is relatively rapid in marine environments [8–10], with laboratory experi-
ments showing degradation rates on the order of days [11–13]. However, in field conditions,
detection of a specific eDNA point source can degrade beyond detection limits in only a few
hours [14,15]. This evidence for rapid eDNA turnover rates suggests that eDNA signatures at a
given location could be highly dynamic over time, particularly for species that are transitory.

In addition to temporal variability, eDNA signatures vary across horizontal space. For
example, Port et al. [16] distinguished distinct communities from eDNA samples separated by
only 60 m in kelp forest ecosystems, and other studies report spatial variation in eDNA signa-
tures on similar spatial scales [17,18]. In contrast, other studies such as O'Donnell et al. [19]
and Lamy et al. [20] detect variation among communities across much larger distances, on the
order of thousands of meters. A key difference in these studies is the latter two examine eDNA
from a dynamic nearshore ecosystem, whereas Port et al. examined eDNA from a highly pro-
tected cove, suggesting that physical oceanographic processes may influence the scale of eDNA
variation.

Although the studies above demonstrate variation of eDNA community signatures across
horizontal space, less is known about variation in marine eDNA signatures across depth. In
fact, most eDNA studies collect samples from consistent depths to control for depth variation
[16,19,21]. The few studies that sample across multiple depths found significant variation in
eDNA signatures. For example, Yamamoto et al. [17] found differences in Japanese jack mack-
erel eDNA concentrations between samples from 1.5 m and 31 m depth. Andruszkiewicz et al.
[22] and Lacoursière-Roussel et al. [23] report similar differences in eDNA signatures between
samples collected at the surface and at depth, suggesting that eDNA can resolve vertical differ-
ences in marine communities. In contrast, Stoeckle et al. [24] did not find eDNA differences
between surface and bottom samples. They note, however, that their pelagic sampling area
supports relatively uniform communities throughout the water column, indicating that their
eDNA results may still reflect the communities present.

Importantly, the scale of depth variation of eDNA signatures in these studies was multiple
tens of meters, yet marine communities can partition along much smaller depth gradients in
nearshore environments [25,26]. In one study, Jeunen et al. [27] found fine-scale depth varia-
tion in eDNA signatures from samples separated by only 4 vertical meters. However, these
samples were taken from a fjord characterized by a strong halocline and very low wave energy,
both of which greatly reduce vertical mixing. It is unclear whether similarly fine-scale vertical
variation in eDNA signatures exists in exposed, nearshore coastal marine ecosystems where
dynamic physical oceanographic processes should promote vertical mixing, potentially
homogenizing eDNA signatures across depth [27].

To better understand the scale of spatial and temporal variation of eDNA signatures, we
examined eDNA vertebrate community composition in a nearshore kelp forest and adjacent
surf zone habitat in Malibu, CA where dynamic physical oceanography may homogenize
eDNA community signatures. Specifically, we test 1) whether eDNA can differentiate among

vertically structured vertebrate communities across a fine-scale depth gradient, 2) whether eDNA signals from similar depths vary among adjacent near-shore habitats, and 3) whether these patterns are stable over time in an exposed coastal marine environment.

## Materials and methods

We conducted our study at Leo Carrillo State Beach, Malibu, California, USA (34.0446˚ N, 118.9407˚ W). We sampled on three successive days (September 24 to September 26 in 2018) to test for temporal stability of spatial variation in eDNA signatures. On each day, we sampled at the highest tide of the mixed semidiurnal tide to minimize the impact of tidal variation on sampling.

To ensure that results reflected variation in spatial sampling, rather than time, we synchronized watches and worked in multiple teams to simultaneously sample from five depths on SCUBA along a vertical transect in a kelp forest ~140 m from shore: 0 m (at the ocean surface), 1 m, 5 m, 9 m, and 10 m (just above the sea floor). We also sampled a sixth station along the shore in the surf zone, where we collected samples approximately 1 m below the water surface where depth was approximately 2 meters. At each location, we collected triplicate seawater samples using one-liter enteral feeding bags (Kendall-Covidien– 702500) following the methods of Curd et al [28]. After returning to shore, we immediately gravity filtered all samples through 0.22 μm Sterivex filters to isolate eDNA [28]. We similarly filtered one liter of distilled water as a negative template control. Upon completion of filtration, we stored the dried filters at -20˚C until extraction 48 hours later. No permits were required for sampling seawater in state waters outside of designated protected areas.

We extracted the DNA from the filters at UCLA using the Qiagen DNEasy Blood and Tissue kit (Qiagen, Valencia, CA, USA). To maximize eDNA recovery, we employed modifications made by Spens et al. [29], adding proteinase K and ATL buffer directly to the filter cartridges before overnight incubation in a rotating incubator at 56˚C. We amplified the extracted eDNA using the *12S* MiFish Universal Teleost (MiFish-U) and MiFish Elasmobranch (MiFish-E) primers with linker modifications for Nextera indices (S1 Table) [30]. Though the primers target teleost fish and elasmobranchs, they can also amplify other vertebrate species such as birds and mammals [30,31]. PCR amplification and library preparation was conducted following the methods of Curd et al. [28] (S1 Appendix). After library preparation, we sequenced the library on a NextSeq at the Technology Center for Genomics & Bioinformatics (University of California, Los Angeles, CA, USA) using Reagent Kit V3 with 30% PhiX added to the sequencing run.

We processed the resulting sequencing data using the *Anacapa Toolkit* (version 1.0) for quality control, amplicon sequence variant parsing, and taxonomic assignment using standard parameters [28]. The *Anacapa Toolkit* sequence quality control and amplicon sequence variant (ASV) parsing module relies on *cutadapt* (version 1.16) [32], *FastX-toolkit* (version 0.0.13) [28], and *DADA2* (version 1.6) [33] as dependencies and the *Anacapa classifier* module relies on *Bowtie2* (version 2.3.5) [34] and a modified version of *BLCA* [35] as dependencies. We processed sequences using the default parameters and assigned taxonomy using two *CRUX*-generated reference databases following the methods of Gold et al. [18]. We first assigned taxonomy using the California Current Large Marine Ecosystem fish specific reference database [36]. Second, we used the *CRUX*-generated *12S* reference database supplemented with California Current Large Marine Ecosystem fish specific references to assign taxonomy using all available *12S* reference barcodes to identify any non-fish taxa following the methods of Gold et al. [36] using a Bayesian cutoff score of 60. Although *CRUX* relies on *ecoPCR* (version 1.0.1) [37], *blastn* (version 2.6.0) [38], and *Entrez-qiime* (version 2.0) [28] as dependencies, we note that Bayesian cutoff scores are

not directly analogous to percent identity from *blastn*. The *BLCA* classifier incorporates alignment metrics, including percent identity and percent overlap, into the underlying Bayesian model which then returns the Bayesian cutoff score metric as a measure of confidence for each taxonomic rank for a given ASV (See [36] and [28] for more detail).

The resulting *Anacapa*-generated taxonomic tables were transferred into R for further processing [36,39]. We then decontaminated the taxonomic tables using methods developed by Kelly et al. [8] and McKnight et al. [40] as implemented in Gold [41], which removes sequences from index hopping and negative controls and conducts a site occupancy model to identify true rare sequences (S2 Appendix). We also manually removed sequences for species with taxonomic assignments for non-marine taxa (e.g. terrestrial mammals) in R. We then merged ASVs by summing reads by assigned taxonomy (e.g. summed all sequences reads from the 7 ASVs that assigned to Garibaldi, *Hypsypops rubicundus*).

Following decontamination, we converted the taxonomic tables into *phyloseq* objects (version 1.30.0) in *R* [42]. We analyzed the eDNA signatures across depth and across nearshore vs. surf zone habitats. We analyzed differences in eDNA across depth using only the nearshore signatures, excluding the samples collected in the adjacent surf zone. To examine species richness across depths, we conducted ANOVA and post-hoc Tukey tests using eDNA read counts (see S2 Table for eDNA read counts) [43]. We then transformed eDNA read counts to eDNA index scores, which better correlates to abundance, following the methods of Kelly et al. [44] (see S3 Table for eDNA Index scores). The eDNA index calculation standardizes eDNA abundance across samples and across taxa. To calculate the eDNA index values, we first calculated the relative abundance of each taxa in each sample. We then divided the relative abundance of each taxa by it's maximum observed abundance across all samples to standardize the read counts per species per sample. This results in an index that ranges from 0 to 1 for each species where a value of 1 corresponds to the sample with the greatest relative abundance observed for that species. Although these eDNA index values allow for direct comparisons of relative abundance within individual species, allowing for direct comparisons in relative abundance by depth, location and/or time, it cannot be used for comparisons among different species. See Kelly et al. [45] for more detail.

To analyze the importance of sampling depth on eDNA vertebrate community composition, we conducted a PERMANOVA test using the *vegan* (version 2.5–6) package in *R* [43]. The PERMANOVA was run using Bray-Curtis dissimilarity and the model eDNA_Index ~ Depth + Day + Replicate. We also ran a multivariate homogeneity of group dispersions test using the betadisper function and Bray-Curtis dissimilarity using *vegan*. We then ran a Mantel Test and non-metric multi-dimensional scaling (NMDS) using *vegan* on Bray-Curtis dissimilarities to assess community composition differences across the depth gradient. We further analyzed vertical depth community composition by generating a gradient forest model using the *gradientForest* package (version 0.1–17) using 500 runs [45]. The environmental variables in the vertical depth gradient forest model included sampling depth, sampling day, and replicate. We then extracted the taxa with the highest model performances and plotted their eDNA index values across depth. Using the broken stick method [46], we defined highest model performance as $\geq 0.35$ $R^2$ importance, as there was a steep drop-off in $R^2$ importance after 0.35.

To analyze differences between the kelp forest and surf zone, we ran Welch t-tests, PERMANOVA and betadisper tests, only including samples taken at 1 m depths from nearshore and surf zone habitats. We used the Bray-Curtis dissimilarity for both tests and the model eDNA_Index ~ Habitat (nearshore vs. surf zone) + Depth + Day + Replicate for the PERMANOVA. We also ran an additional gradient forest model with the environmental variables sampling depth, nearshore vs. surf zone, sampling day, and replicate for eDNA index scores from all stations. We then extracted the top performing taxa and plotted their eDNA index

distributions, defining highest model performance as $\geq 0.40$ $R^2$ importance using the broken stick method, as there was a steep drop-off in $R^2$ importance after 0.40. Because of the additional nearshore vs. surf zone variable, the $R^2$ importance values were higher in this model, resulting in a different threshold $R^2$ value than the one in the depth gradient forest model.

To test whether vertical and horizontal variation in eDNA signatures were consistent over time, we compared species richness across sampling days in an ANOVA framework, looking at both total community diversity as well as the eDNA index abundances. The linear models used for the eDNA index ANOVA tests were eDNA_Index ~ Station + Day + Station:Day.

## Results

We generated a total of 23,504,223 sequence reads that passed filter from the NextSeq run. After decontamination, we recovered 21,231,865 reads and 980 ASVs representing 48 families, 71 genera and 71 species, with an average of 5.3 ASVs per taxa. Species detected included teleost fish (n = 57), elasmobranchs (n = 8), marine mammals (n = 2) and birds (n = 6) (S4 Table). Most teleost fish and elasmobranchs were demersal (rocky reef or sandy bottom species) (n = 56; 86%), while the others were pelagic (n = 9; 14%) (S4 Table) [47]. Of the six bird species, three were seabirds (Western gull *Larus occidentalis*, Brown pelican *Pelecanus occidentalis* and Pelagic cormorant *Urile pelagicus*) and three were terrestrial birds (American golden plover *Pluvialis dominica*, Band-backed wren *Campylorhynchus zonatus*, and Ruby-crowned kinglet *Regulus calendula*). The two marine mammal species were the Bottlenose dolphin (*Tursiops truncates*) and California sea lion (*Zalophus californianus*).

### Vertical depth comparisons

**Species richness.** Species richness differed significantly across depth, with shallow sampling stations having lower species richness than deeper stations (ANOVA; p<0.001; Fig 1). Specifically, the 0m and 1m species richness values were significantly different from the 9m (both p = 0.001) and 10m (p = 0.001 and p = 0.002, respectively) species richness values. Mean species richness for the 0 m and 1 m sites were 31.4 and 32.1, respectively, while the 5 m, 9 m, and 10 m sites were 38.2, 45.7 and 44.8, respectively.

**Community composition.** Community composition differed significantly across depth (PERMANOVA; p = 0.001). Depth accounted for most of the variation in community composition ($R^2$ = 0.16), followed by sampling day ($R^2$ = 0.14) and bottle replicate ($R^2$ = 0.03) (S1 Fig).

Pairwise PERMANOVA comparisons indicated that the 5 m, 9 m and 10 m stations were all significantly different from the 0 m and 1 m stations (all p<0.05). The 5 m station was significantly different from the 10 m station (p<0.05) but not the 9 m station (p = 0.14). Group dispersions also differed between the 1 m and 5 m stations (p = 0.02).

Mantel tests indicate that community composition significantly correlated with depth (Mantel statistic r = 0.405, p = 0.001). Further support for this result comes from the NMDS plots, which show that communities closer in vertical distance more closely resemble each other than communities separated by greater vertical distance (Fig 2).

Of the environmental variables in the gradient forest model, depth had the highest accuracy importance (0.010) and $R^2$ importance (0.111) values, followed by sampling day (accuracy importance: 0.010; $R^2$ importance: 0.109) and replicate (accuracy importance: -0.001; $R^2$ importance: 0.006) (S2 Fig). There were 14 taxa characterized by model performances of $R^2$ importance > 0.35 (S3 Fig). Pacific sardine *Sardinops sagax*, Topsmelt silverside *Atherinops affinis* and California grunion *Leuresthes tenuis* were most abundant the shallow stations (0 m and 1 m station). The remaining eleven taxa (Yellowfin drum *Umbrina roncador*, Barred sand bass *Paralabrax nebulifer*, California anchovy *Engraulis mordax*, Kelp bass *Paralabrax clathratus*,

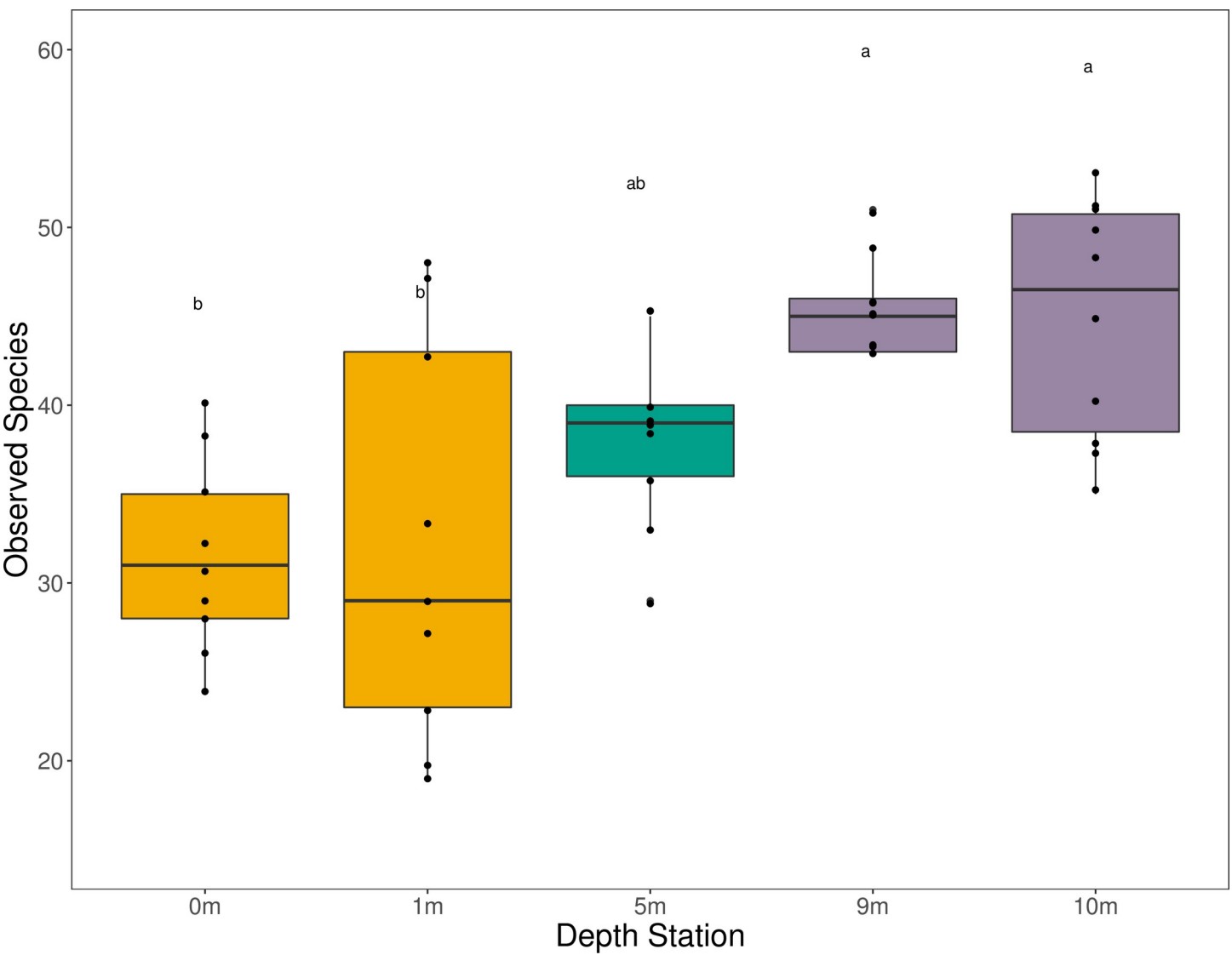

**Fig 1. Boxplot of mean species richness (± standard error) across depth stations.** The deep stations had higher species richness than the shallow stations.
Figure includes only the nearshore samples and does not include the surf zone samples. The colors represent relative depths of the stations: Yellow is shallow (0 m and 1 m), green is mid-water (5 m), and purple is deep (9 m and 10 m).

Speckled sanddab *Citharichthys stigmaeus*, California kingcroaker *Menticirrhus undulatus*, the surfperches of Family Embiotocidae, Queen croaker *Seriphus politus*, California sheephead *Semi-cossyphus pulcher*, Pacific angelshark *Squatina californica* and clinids of the Genus *Gibbonsia*) were most abundant in the mid and deeper stations (5 m, 9 m and 10 m) (Fig 3).

Overall, of the 67 teleost and elasmobranch taxa, 16 (24%) were most abundant in the shallow stations and 50 (76%) were most abundant in the mid and deep stations (S5 Table). One species, Gray smoothhound *Mustelus californicus*, was only found in the surf zone station.

## Nearshore vs. surf zone analysis

**Species richness.** The surf zone station had a mean of 44.1 taxa per sample compared to 32.1 per sample for the corresponding 1 m nearshore station. Species richness differed between the 1 m surf zone and 1 m nearshore station (Welch two sample t-test: p = 0.02).

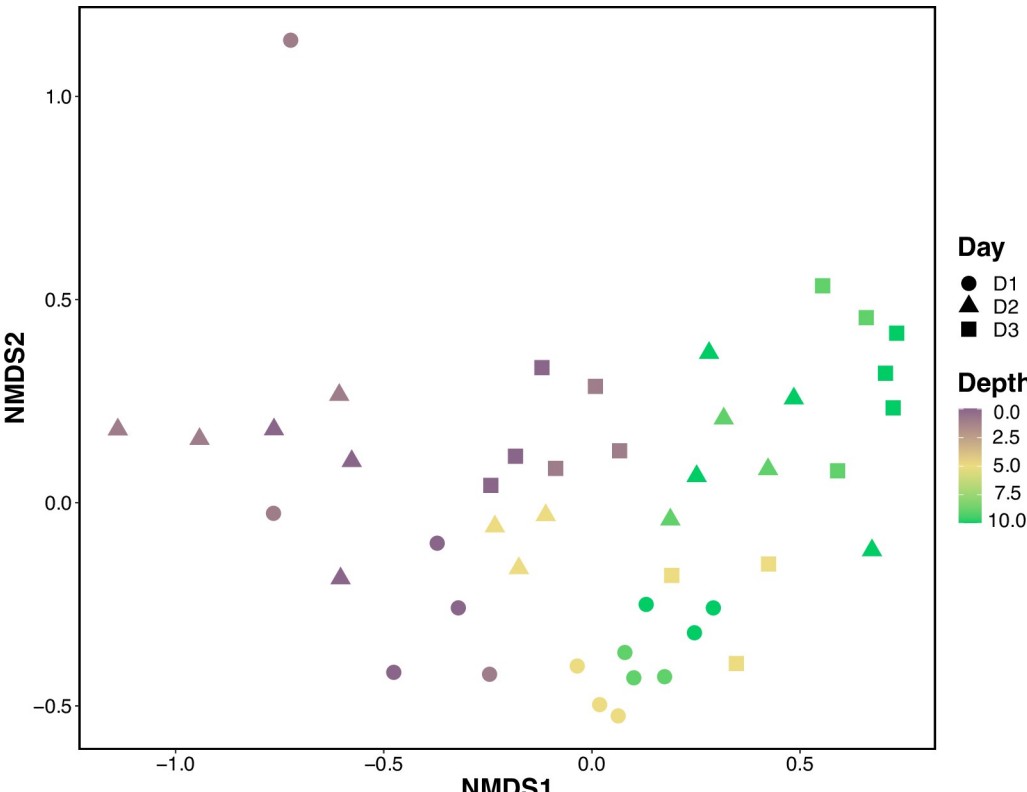

**Fig 2. NMDS of species assemblages across depth (m) using Bray-Curtis dissimilarity.** Differences in community composition increased with greater distance between depth stations (stress = 0.178). 0 m represents the surface and 10 m represents the sea floor.

**Community composition.** Marine vertebrate community composition differed significantly between the 1 m surf zone and 1 m nearshore stations (PERMANOVA: p = 0.001; S4 Fig). The 1 m surf zone station also had significantly lower group dispersions than the 1 m nearshore station (p = 0.002).

In the nearshore vs. surf zone gradient forest model, depth still had the highest accuracy importance (0.011) and $R^2$ importance (0.122) values (S5 Fig). Nearshore vs. surf zone designations had the second highest accuracy importance (0.006) and $R^2$ importance (0.060) values, followed by sampling day (accuracy importance: 0.006; $R^2$ importance: 0.055) and replicate (accuracy importance: -0.001; $R^2$ importance: 0.005) (S5 Fig). There were seven taxa with performances in the model with > 0.40 $R^2$ importance values (S6 Fig). Five taxa (California king-croaker *Menticirrhus undulatus*, surfperches of Family Embiotocidae, Yellowfin drum *Umbrina roncador*, Queen croaker *Seriphus politus* and Zebra perch sea-chub *Hermosilla azurea*) were most abundant in the surf zone station (Fig 4). Barred sand bass *Paralabrax nebulifer* and Kelp bass *Paralabrax clathratus* were most abundant in the deep nearshore stations (9m and 10 m) (Fig 4).

## Temporal comparisons

**Species richness.** Species richness did not differ across sampling days (ANOVA: p = 0.195). Mean species richness was 37.8 on Day 1, 38.1 on Day 2 and 42.6 on Day 3.

**Community composition.** There were fourteen taxa that varied strongly in the gradient model with > 0.35 $R^2$ importance values (S3 Fig). Six taxa (Yellowfin drum *Umbrina roncador*,

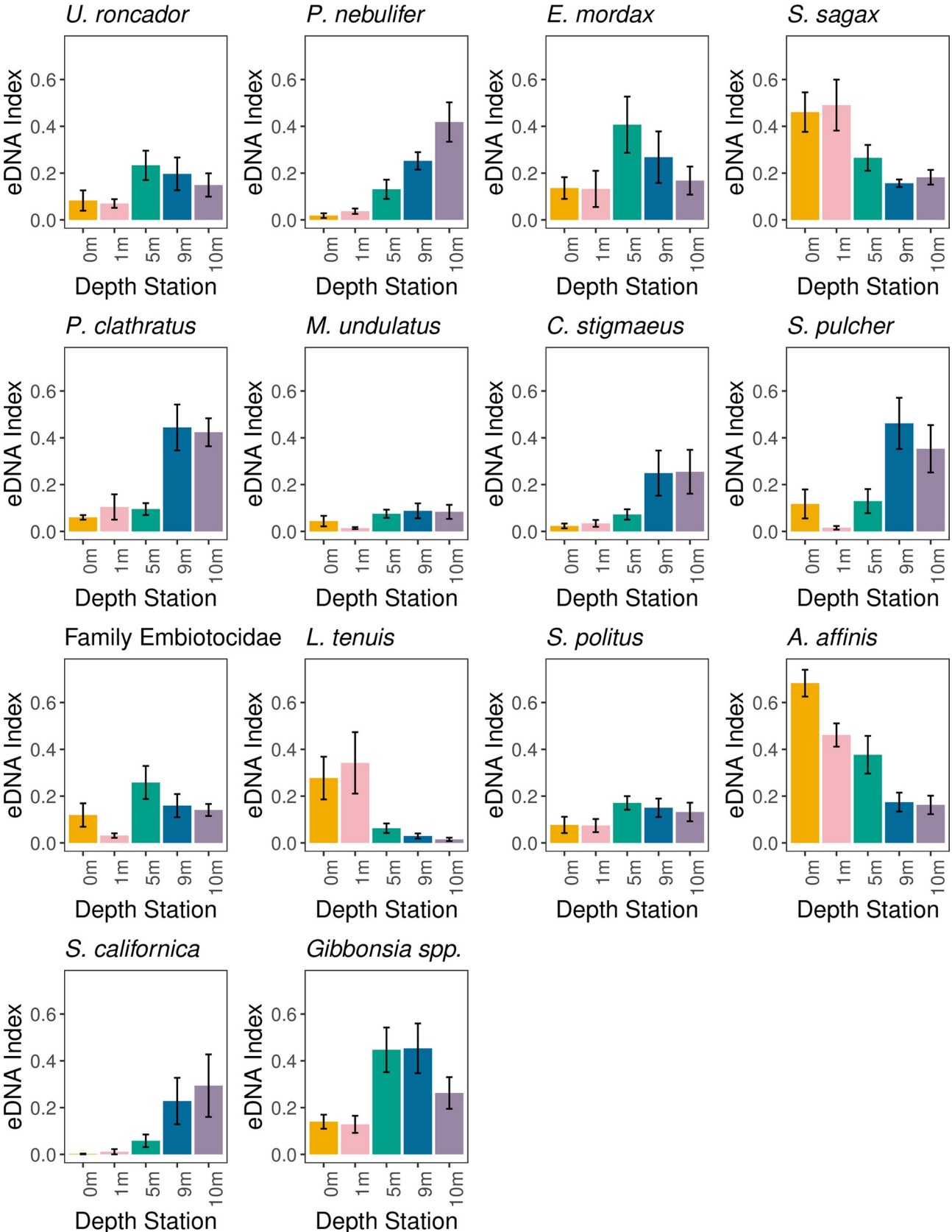

**Fig 3. eDNA Index abundance depth distributions for the top predictor taxa.** eDNA Index mean and standard error (+/- 1SE) for taxa with greater than 0.35 $R^2$ importance in the gradient forest model. See S1 and S2 Appendices for assigning. Family Embiotocidae and Genus *Gibbonsia*.

California anchovy *Engraulis mordax*, California kingcroaker *Menticirrhus undulatus*, surfperches of Family Embiotocidae, Queen croaker *Seriphus politus*, and clinids of Genus *Gibbonsia*) were most abundant on Day 1, three taxa (Pacific sardine *Sardinops sagax*, Topsmelt silverside *Atherinops affinis*, California grunion *Leuresthes tenuis*) were most abundant on Day 2, and four taxa (Barred sand bass *Paralabrax nebulifer*, Speckled sanddab *Citharichthys stigmaeus*, California sheephead *Semicossyphus pulcher*, Pacific angelshark *Squatina californica*) were most abundant on Day 3 (S6 Table). Kelp bass *Paralabrax clathratus* was most abundant in Day 2 and Day 3 (S6 Table).

Of the 67 teleost and elasmobranch taxa, 46 (69%) had depth distribution patterns that did not vary across the three days ($p > 0.05$; see S7 Table for p-values). Interestingly, 37 of these taxa (80%) are characterized as nonmigratory and non-transitory species (S7 Table). Of the 21 taxa with variable depth distributions ($p < 0.05$), 10 (48%) of the taxa are transitory or migratory (S7 Table).

## Discussion

Results of eDNA surveys in a dynamic, California coastal ecosystem demonstrate fine-scale vertical and horizontal variation in marine vertebrate communities. Differences in vertebrate

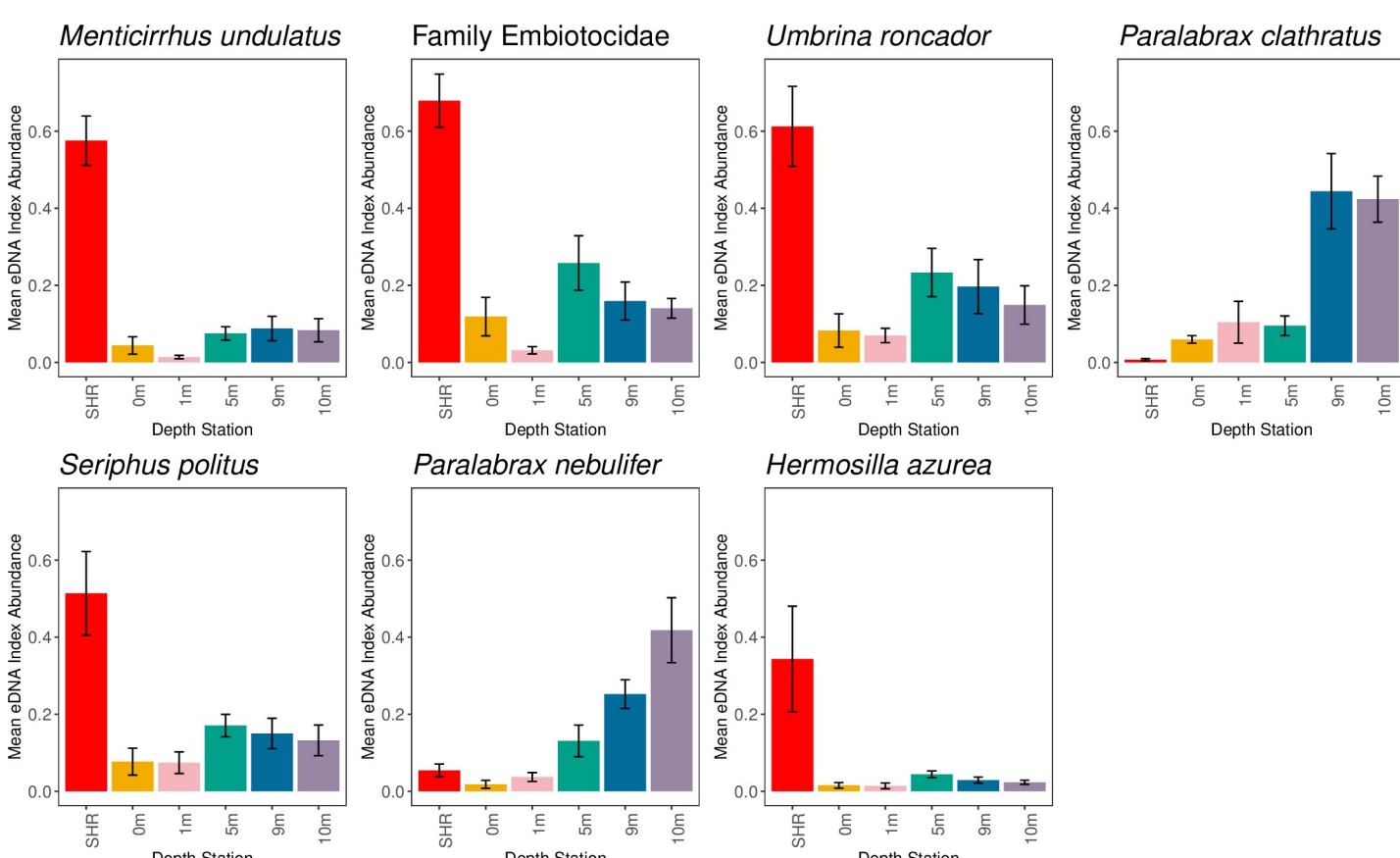

**Fig 4. The top predictor taxa eDNA index abundance nearshore and surf zone distributions.** eDNA Index mean and standard error (+/- 1SE) for the taxa with greater than 0.40 $R^2$ importance in nearshore vs. surf zone gradient forest model. See S1 and S2 Appendices for assigning Family Embiotocidae.

eDNA signatures across a 10 m depth gradient largely reflect species-specific variation in microhabitat depth preferences, particularly within fishes [48]. Similarly, eDNA discriminated between nearshore and surf zone communities with patterns also reflecting known ecological differences among fishes. While eDNA signatures varied across time, signatures of resident taxa were relatively stable. Combined, these results underscore the sensitivity of eDNA to discriminate ecologically relevant vertical, horizontal, and temporal variation.

Other studies report depth variation in eDNA over larger depth ranges [17,22], or similar scales in marine ecosystems with limited vertical mixing [25]. Studies also report horizontal partitioning on larger spatial scales along exposed coastlines [18] or on similar spatial scales in protected coves. Our results are unique in that eDNA discerned fine-scale vertical and horizontal variation in marine vertebrate communities in a dynamic, exposed coastal ecosystem, highlighting the ability of eDNA to provide a highly localized snapshot of marine diversity across depth and habitat type.

## eDNA distinguishes marine communities by depth

Previous eDNA studies report differences in eDNA signatures across depth [17,22,23,27], but these differences were observed across large depth ranges across tens of meters. The finest depth partitioning reported in an eDNA study, only 4 m [27], occurred in a unique marine ecosystem with a pronounced pyncocline driven by a 3–4 ˚C temperature gradient and 20–30 PSU salinity gradient [26], values that greatly exceed those observed in Southern California [49]. In such a stratified water column, in an ecosystem with limited wave energy, vertical movement of eDNA would be limited to diffusion, or potentially other processes such as biogenic vertical mixing [50,51], allowing for fine-scale variation in eDNA signatures.

While results of this study also differentiated eDNA community signatures over 4–5 m, our study site was a wave-exposed coastal environment, where currents and wave energy should facilitate vertical water mixing. Over the three sampling days of this study, daily tidal fluctuations exceeded 1 m (0.15 m low tide, 1.5 m high tide), swell height ranged from 0.23–0.43 m, and surf height ranged from 0.3–0.9 m, providing water movement and wave energy to facilitate mixing. Although we sampled at the end of the boreal summer, when the nearshore Southern California summer thermocline is typically most pronounced and within a few meters of the ocean surface [52,53], observed surface temperatures were typical for the time of year (19.2–19.4˚C) and neither divers or dive computers observed a thermocline that would limit vertical mixing. As such, the community variation in eDNA signatures across a depth gradient of 4–5 m cannot be attributed to water stratification and limited mixing.

Instead, the composition of vertebrate communities detected with eDNA corresponded to species-specific depth preferences [47,48]. Fifty-one (77%) of the taxa had eDNA depth distribution patterns that match their known depth preferences (pelagic vs. demersal). This result held true particularly for 13 of the 14 highest performing taxa ($R^2 > 0.35$) in the gradient forest model. For example, the coastal-pelagic species Pacific sardine *Sardinops sagax*, Topsmelt silverside *Atherinops affinis* and California grunion *Leuresthes tenuis* were most abundant in the 0 m and 1 m stations [48,54–56]. Ten demersal taxa (Barred sand bass *Paralabrax nebulifer*, clinids of the Genus *Gibbonsia*, Yellowfin drum *Umbrina roncador*, Speckled sanddab *Citharichthys stigmaeus*, California kingcroaker *Menticirrhus undulatus*, surfperches of the Family Embiotocidae, Queen croaker *Seriphus politus*, Kelp bass *Paralabrax clathratus*, California sheephead *Semicossyphus pulcher* and Pacific angelshark *Squatina californica*) were most abundant in the mid and deeper stations [48,57]. The correspondence in species-specific eDNA relative abundances and the preferred microhabitats of these species strongly suggests that eDNA is recovering ecologically informative depth variation in marine communities,

adding to a growing list of studies highlighting the ability of eDNA to accurately discriminate fine-scale habitat partitioning in nearshore marine environments [48,57,58].

Species at depth greatly outnumbered species at the surface (≤1 m), with the 0 m and 1 m stations having, on average, 13 fewer taxa than the deeper stations (5 m, 9 m, 10 m). While the partitioning of shallow and deeper water vertebrate communities broadly conformed to known habitat preferences, taphonomic processes may also reduce the detection of surface-dwelling taxa. Solar radiation, particularly ultraviolet (UV) light, is highest in surface waters and may increase DNA denaturation and degradation [59–63]. Similarly, surface water temperatures are warmer, potentially increasing microbial and enzymatic activity, accelerating eDNA degradation [61,64]. Conversely, settlement processes could also elevate eDNA community diversity at depth [65]. Common sources of eDNA (e.g. feces) are often too large to remain suspended, potentially inflating eDNA community diversity at depth as these particles sink [66–68]. In our study, some pelagic, surface-dwelling species were found in the deep stations, possibly due to such settlement processes. For example, Topsmelt silverside *Atherinops affinis*, California grunion *Leuresthes tenuis* and Pacific sardine *Sardinops sagax* were pelagic species found in the 9 m and 10 m stations. Though these three species were found in the deeper stations, they had low abundances in these stations and had much higher abundances in the shallow stations (Fig 3). These relative abundances reflect their habitat preferences, which suggests that, because community composition largely matches microhabitat differences, the impacts of settlement processes are likely relatively minor.

## eDNA fate and transport is likely limited

The strong depth gradient in eDNA signatures strongly suggests limited persistence and transport of eDNA, even in dynamic nearshore coastal environments. However, previous laboratory experiments and modeling studies indicate marine eDNA degradation rates on the scale of multiple days, with potential transport distances of hundreds of meters to kilometers [9,13,17,19]. One possible explanation to reconcile this apparent contradiction is high shedding rates of endogenous eDNA. If local sources of eDNA generation are high and continuous, recently generated eDNA should dominate signatures [8], as eDNA from more distant sources would have much lower concentrations due to diffusion and degradation processes during transport. Controlled field and lab experiments are needed to explicitly test this hypothesis.

## eDNA recovers nearshore and surf zone communities

Surf zone and nearshore rocky reef community eDNA signatures varied significantly, and was comparable to variation observed by O'Donnell et al. [19] over kilometers in a dynamic coastal ecosystem. The PERMANOVA and low group dispersions for the surf zone station both indicate that the surf zone community is distinct from the nearshore community. Importantly, four of the five taxa in the nearshore vs. surf zone gradient forest model that were most abundant in the surf zone station (California kingcroaker *Menticirrhus undulatus*, surfperches of Family Embiotocidae, Yellowfin drum *Umbrina roncador*, and Queen croaker *Seriphus politus)* are all associated with surf zone habitats, providing further evidence that eDNA accurately recovers distinct communities across adjacent habitat types (S4 Table) [48,57]. Combined with the above, these results indicate that eDNA can capture variation in nearshore marine ecosystems, both vertically and across habitats, highlighting that eDNA is a localized snapshot of marine diversity.

## eDNA signatures vary across time

The PERMANOVA and gradient forest analyses indicate eDNA signatures varied daily, similar to previous *in situ* eDNA studies [8–10]. Importantly, much of the temporal variance

appears to result from species behavioral patterns. Of the 67 teleost and elasmobranch species, 47 (70%) had temporal variation in eDNA depth distributions that matched their known behavior. Twenty-one (31%) taxa had significantly variable depth distributions (S7 Table). Of these 21 taxa, ten (48%) are transitory or migratory, including species such as Queen croaker *Seriphus politus*, Pacific sardine *Sardinops sagax*, and the Pacific anchovy *Engraulis mordax* (S7 Table). These mobile species transit in and out of kelp forest ecosystems [48,55], potentially contributing the observed daily temporal variation in eDNA signatures. Furthermore, the relative abundances of less mobile taxa were fairly consistent over time at different depths. Forty-six (69%) of species exhibited consistent eDNA depth distributions across the three days. Of these taxa with consistent depth distributions across the three days, 37 (80%) are nonmigratory and relatively stationary species, such as Speckled sanddab *Citharichthys stigmaeus*, California sheephead *Semicossyphus pulcher* the Barred sand bass *Paralabrax nebulifer* (S7 Table). Together, the majority of species exhibited depth variation were consistent with their behavior, indicating eDNA is likely reflecting behavioral patterns of marine vertebrates within the ecosystems they inhabit (S7 Table).

However, there were taxa whose eDNA temporal variation did not match their known behavior. For example, Kelp bass *Paralabrax clathratus*, a non-migratory, less mobile species, had highly variable eDNA abundances across days, while Chub mackerel *Scomber japonicus*, a transitory species, had consistent abundances. It is unknown whether this discrepancy indicates an incomplete understanding of the ecology of these fishes, variability in generation or degradation rates among these species, or whether physical oceanographic processes obscure potential eDNA-derived behavior patterns (e.g. algae dwelling species eDNA trapped within the boundary layer). Regardless, the broad concordance of eDNA distributions and species ecologies underscores its ability to accurately recover marine communities over space, depth and time, opening the door to future applications of eDNA to better understand the behavior of marine organisms.

## Conclusions

Our study demonstrates the power of eDNA to distinguish unique vertebrate community microhabitats, both across depth at a single location, and across horizontally distinct communities. Patterns in spatial partitioning were relatively stable despite sampling a dynamic, nearshore marine environment, and reflected ecological differences in vertebrate communities. This consistency provides confidence for the application of eDNA methods in coastal biodiversity assessments. For example, eDNA can help us better understand species habitat distributions and how these distributions change over time with increased global change. However, our results also highlight the incredible sensitivity of eDNA metabarcoding approaches and suggest eDNA signatures only integrate biodiversity information across short time periods and small depth ranges.

These results underscore the importance of consistent sampling depth in marine eDNA studies, as variation in sampling depth could impact results. Moreover, for studies that seek to maximize sampling of local biodiversity through eDNA, efforts may need to incorporate sampling across horizontal space, depth, and time, as one depth and one location may not accurately reflect the full scope of local biodiversity of a given kelp forest. Fortunately, given the ease and cost-effective nature of eDNA sampling, such sampling efforts are not cost prohibitive, and will help us better document and monitor changes in coastal marine biodiversity.

## Supporting information

**S1 Fig. Apportioned variance plot of the three variables in the PERMANOVA model.**
Depth accounted for 16% variance (p = 0.001), collected date accounted for 14% variance

(p = 0.001) and replicate accounted for 3% variance (p = 0.426).
(TIFF)

**S2 Fig. Importance of variables in the depth gradient forest model.** Sampling depth had the highest accuracy importance and $R^2$ weighted importance in the depth gradient model.
(TIFF)

**S3 Fig. Performance of species in the depth gradient forest model.** There were fourteen top predictor species with $R^2$ values greater than 0.35 in the gradient forest model.
(TIFF)

**S4 Fig. NMDS for shore vs. 1 m stations.** Surf zone community composition differs from the nearshore community composition. NMDS stress is 0.082.
(TIFF)

**S5 Fig. Importance of variables in the nearshore vs. surf zone gradient forest model.** Sampling depth had the highest accuracy importance and $R^2$ weighted importance in the nearshore vs. surf zone space gradient forest model.
(TIFF)

**S6 Fig. Performance of species in the nearshore vs. surf zone space gradient forest model.** There were seven top predictor species with $R^2$ values greater than 0.40 in the gradient forest model.
(TIFF)

**S1 Table. MiFish-U and MiFish-E primer sequences.**
(XLSX)

**S2 Table. Taxonomy table with read counts after decontamination.**
(XLSX)

**S3 Table. Taxonomy table with eDNA Index values after decontamination.**
(XLSX)

**S4 Table. Species list.**
(XLSX)

**S5 Table. eDNA abundance patterns (shallow vs. deep stations) for all teleost and elasmobranch taxa.**
(XLSX)

**S6 Table. Mean eDNA Index across days for top predictor taxa in the gradient forest model.**
(XLSX)

**S7 Table. Ecology, behavior and depth distribution variability across days.** Table includes p-values for the Day:Station interaction of the ANOVA for the linear model of eDNA index abundance.
(XLSX)

**S1 Appendix.**
(PDF)

**S2 Appendix.**
(PDF)

## Acknowledgments

We thank Camille Gaynus, Nury Molina, Eric Caldera, Erick Zerecero, and Elizabeth Reid-Wainscoat for their assistance in eDNA sample collection, and thank the Barber lab undergraduates Markus Min, McKenzie Koch, Bridget Foy, Beverly Shih, Nikita Sridhar, and Cristopher Ruano for their help with laboratory work.

## Author Contributions

**Conceptualization:** Keira Monuki, Paul H. Barber, Zachary Gold.

**Data curation:** Keira Monuki, Zachary Gold.

**Formal analysis:** Keira Monuki, Zachary Gold.

**Funding acquisition:** Keira Monuki, Paul H. Barber, Zachary Gold.

**Investigation:** Keira Monuki, Zachary Gold.

**Methodology:** Keira Monuki, Paul H. Barber, Zachary Gold.

**Project administration:** Keira Monuki, Paul H. Barber, Zachary Gold.

**Resources:** Paul H. Barber, Zachary Gold.

**Software:** Keira Monuki, Zachary Gold.

**Supervision:** Paul H. Barber, Zachary Gold.

**Validation:** Paul H. Barber, Zachary Gold.

**Visualization:** Keira Monuki, Zachary Gold.

**Writing – original draft:** Keira Monuki.

**Writing – review & editing:** Keira Monuki, Paul H. Barber, Zachary Gold.

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
