## [Decision Letter · Decision Letter 0]

27 Aug 2021

PONE-D-21-17590

eDNA Captures Microhabitat Partitioning in a Kelp Forest Ecosystem

PLOS ONE

Dear Dr. Monuki,

Thank you for submitting your manuscript to PLOS ONE. After careful consideration, we feel that it has merit but does not fully meet PLOS ONE’s publication criteria as it currently stands. Therefore, we invite you to submit a revised version of the manuscript that addresses the points raised during the review process.

I got the recommendations and comments from an expert reviewer on the field, because it was very difficult to find more reviewers. The  reviewer agree that the manuscript is technically sound and the data support the conclusions. However, lack of the explanation in Methods and Results sections were suggested by the reviewer, and I totally share their comments. Therefore, I can invite you to submit a revised version of the manuscript that addresses the points raised by the reviewer, and may re-review by the reviewer and additional reviewers.

We look forward to receiving your revised manuscript.

Kind regards,

Hideyuki Doi

Academic Editor

PLOS ONE

Journal Requirements:

5. We note that you have referenced (ie. Bewick et al. [5]) which has currently not yet been accepted for publication. Please remove this from your References and amend this to state in the body of your manuscript: (ie “Bewick et al. [Unpublished]”) as detailed online in our guide for authors

Additional Editor Comments:

I got the recommendations and comments from an expert reviewer on the field, because it was very difficult to find more reviewers. The reviewer agree that the manuscript is technically sound and the data support the conclusions. However, lack of the explanation in Methods and Results sections were suggested by the reviewer, and I totally share their comments. Therefore, I can invite you to submit a revised version of the manuscript that addresses the points raised by the reviewer, and may re-review by the reviewer and additional reviewers.

Reviewers' comments:

Reviewer's Responses to Questions

**Comments to the Author**

1. Is the manuscript technically sound, and do the data support the conclusions?

Reviewer #1: Partly

2. Has the statistical analysis been performed appropriately and rigorously? 

Reviewer #1: Yes

3. Have the authors made all data underlying the findings in their manuscript fully available?

Reviewer #1: Yes

4. Is the manuscript presented in an intelligible fashion and written in standard English?

Reviewer #1: Yes

5. Review Comments to the Author

Reviewer #1: This manuscript reports on vertebrate eDNA by depth in a kelp forest California. At a site 140 meters offshore, water sample were collected daily for 3 days at five depths from surface to bottom--0, 1, 5, 9,and 10m (bottom), plus at an adjacent surf zone site. This is a well done study that adds useful information on how eDNA signals differ by depth in this ecosystem, which is highly relevant to designing and interpreting eDNA surveys of marine animals. This is one of few studies to date looking at eDNA differences by depth.

My main suggestions for improvement are that the study design and methods could be more clearly described, some limitations need to be addressed, and some of data are over-interpreted.

Specific comments.

Line 4. I'm not sure what "microhabitats" are. The main finding is fish species richness and community composition differs from surface (0-1m) and bottom (4-5 m). I think a title with "depth" in it would be more informative.

line 25. Is the surf zone part of the kelp forest? If not you need to re-word this.

line 27. Only 3 marine birds and 2 marine mammals were detected, this does not qualify as "a broad-range of marine vertebrates". With so few species, findings on birds and mammals are of uncertain signficance, would take out of abstract.

line 29. "across 4-5m depth" is unclear, how about "between surface and depth"

line 34. Unlike depth and surf zone/kelp forest findings, for temporal variation, you don't have evidence that this reflects variation in local marine biodiversity. It could be due to variable transport or degradation of eDNA. I think this part of conclusion should be omitted.

Line 51. Regarding traditional methods, most marine surveys are not SCUBA-based, you include references to trawl and sound based methods.

line 57. Nicely said regarding methodological biases.

line 76. "vertically" and "depth" are redundant, just say one or the other.

line 81. Re eDNA by depth studies, suggest include ICES J 2020 Stoeckle et al which did not find eDNA differences by depth in coastal ocean up to 30m depth.

Line 121. Regarding methods, please include here or in Appendix S1 what was extracted DNA volume, so reader can know what proportion is represented by 1ul DNA input. Also it's not clear when the MiFish-U and MiFish-E amplifications were combined--after sequencing?

line 149. The paragraph in Supplemental Methods S1 which describes how reads were converted to indices belongs in the main text. Also a sentence stating the limitation of this approach, which is that you can't compare relative abundance across species, only relative abundance within species.

line 151. I may have missed this, I didn't understand the point of analyzing the more frequently occurring species separately. To my eye there is no clear break 80% in Table S1. I see on line 407 that the core group explained variance better, but I didn't see what the point of that is, the direction of difference is the same. If you're going to exclude data, you need a better rationale than that it makes findings look better.

Line 174. I think it would be more accurate to say differences between kelp forest and surf zone, not "differences across horizontal space".

line 189. It would help to better understand how ID works. There were 980 ASVs, and a total of 71 vertebrate species. What happened to all the ASVs? Is there a lumping threshold? What about likely contaminants such as human DNA? In Supplemental Methods S2 it refers to IDs with "55-59% confidence scores" as being "high confidence". That doesn't sound like a good match to me, percent identity might be more informative?

Line 202. This is a repeat of sentence on line 192.

Line 211. I can guess but what do the box plot lines and whiskers represent?

Line 211. I think you should include that the surf zone data is not included in figure. It's slightly misleading since the surf zone data was used to calculate the indices.

Line 215. Can't you summarize p-values somehow in text, instead of referring reader to Supplementary Material?

Also please be specific if sequences were matched to a local fish database, or to whatever is in GenBank.

Appendix S1 line 10. I don't understand how triplicate amplification helps "to account for PCR artifacts".

6. PLOS authors have the option to publish the peer review history of their article (what does this mean?). If published, this will include your full peer review and any attached files.

Reviewer #1: No

---

## [Author Response · Author response to Decision Letter 0]

10 Oct 2021

Reviewer #1: This manuscript reports on vertebrate eDNA by depth in a kelp forest California. At a site 140 meters offshore, water sample were collected daily for 3 days at five depths from surface to bottom--0, 1, 5, 9,and 10m (bottom), plus at an adjacent surf zone site. This is a well done study that adds useful information on how eDNA signals differ by depth in this ecosystem, which is highly relevant to designing and interpreting eDNA surveys of marine animals. This is one of few studies to date looking at eDNA differences by depth.

My main suggestions for improvement are that the study design and methods could be more clearly described, some limitations need to be addressed, and some of data are over-interpreted.

We thank you for your review and for your kind words about the importance of our study. We appreciate and agree with your suggestions and have addressed the specific comments below. The main change is the removal of the highest occurring species analyses and the inclusion of analyses for all species.

Specific comments.

Line 4. I'm not sure what "microhabitats" are. The main finding is fish species richness and community composition differs from surface (0-1m) and bottom (4-5 m). I think a title with "depth" in it would be more informative.

We agreed with this suggestion and changed the title to “eDNA Captures Depth Partitioning in a Kelp Forest Ecosystem” (Line 4)

line 25. Is the surf zone part of the kelp forest? If not you need to re-word this.

The surf zone is not part of the kelp forest. We clarified this distinction in the sentence “[…] eDNA signatures across depth (0 m to 10 m) and horizontal space (nearshore kelp forest and surf zone) over three successive days in Southern California” (New Line 27)

line 27. Only 3 marine birds and 2 marine mammals were detected, this does not qualify as "a broad-range of marine vertebrates". With so few species, findings on birds and mammals are of uncertain signficance, would take out of abstract.

Thank you for the suggestion. We changed the sentence to “Across a broad range of teleost fish and elasmobranchs […]” (New Line 28)

line 29. "across 4-5m depth" is unclear, how about "between surface and depth"

We changed the sentence to reflect this suggestion “[…] results showed significant variation in species richness and community assemblages between surface and depth […]” (New Line 30)

line 34. Unlike depth and surf zone/kelp forest findings, for temporal variation, you don't have evidence that this reflects variation in local marine biodiversity. It could be due to variable transport or degradation of eDNA. I think this part of conclusion should be omitted.

We removed the previous sentence “Patterns of microhabitat partitioning in eDNA signatures across space and time were largely consistent with known habitat preferences and species behavior” (Old Line 32). We added the sentences “Community assemblages between nearshore and surf zone sampling stations at the same depth were also significantly different and consistent with known habitat preferences. Assemblages also varied across three sampling days, but 69% of habitat preferences remained consistent.” (New Line 31)

Line 51. Regarding traditional methods, most marine surveys are not SCUBA-based, you include references to trawl and sound based methods.

Thank you for this comment. We changed the wording to “traditional surveys” (New Line 51)

line 57. Nicely said regarding methodological biases.

Thank you. 

line 76. "vertically" and "depth" are redundant, just say one or the other.

We removed “vertically” (New Line 77)

line 81. Re eDNA by depth studies, suggest include ICES J 2020 Stoeckle et al which did not find eDNA differences by depth in coastal ocean up to 30m depth.

We thank you for suggesting this study. We included a brief discussion of their findings in the Introduction: “In contrast, Stoeckle et al. [23] did not find eDNA differences between surface and bottom samples. They note, however, that their sampling area supports relatively uniform communities throughout the water column, indicating that their eDNA results may still reflect the communities present.” (New Line 84)

Line 121. Regarding methods, please include here or in Appendix S1 what was extracted DNA volume, so reader can know what proportion is represented by 1ul DNA input. Also it's not clear when the MiFish-U and MiFish-E amplifications were combined--after sequencing?

Thank you for these comments. We added a sentence to the beginning of S1 Appendix “The extracted DNA volume for each sample was 100 μL.” (New Line 1). To clarify when the MiFish-U and MiFish-E amplifications were combined, we added a sentence in S1 Appendix: “We indexed the MiFish-U and MiFish-E PCR products from the same sample with the same index, except for one sample with different indices for the MiFish-U and MiFish-E PCR products (sample Day2_10m_A).” (New Line 19). We also added “[…] pooled even copy numbers for each marker (MiFish-U and MiFish-E)” for clarity. (New Line 29) 

line 149. The paragraph in Supplemental Methods S1 which describes how reads were converted to indices belongs in the main text. Also a sentence stating the limitation of this approach, which is that you can't compare relative abundance across species, only relative abundance within species.

Thank you for these suggestions. We added S1 Supplemental Methods to main text (New Line 169) and removed them from the Supplemental Methods. Also added a sentence on the limitations of using eDNA index (New Line 175): “The eDNA index calculation standardizes eDNA abundance across samples and across taxa. To calculate the eDNA index values, we first calculated the relative abundance of each taxa in each sample. We then divided the relative abundance of each taxa by it’s maximum observed abundance across all samples to standardize the read counts per species per sample. This results in an index that ranges from 0 to 1 for each species where a value of 1 corresponds to the sample with the greatest relative abundance observed for that species. We note that this transformation does not allow for direct comparisons of relative abundance between species only within species. See Kelly et al. [45] for more detail.” (New Lines 169-178)

line 151. I may have missed this, I didn't understand the point of analyzing the more frequently occurring species separately. To my eye there is no clear break 80% in Table S1. I see on line 407 that the core group explained variance better, but I didn't see what the point of that is, the direction of difference is the same. If you're going to exclude data, you need a better rationale than that it makes findings look better.

We agree with this suggestion. Analyses of frequently occurring species were removed from the Results and Discussion sections, and analyses were added for all species. In the Results, we added sentences “Overall, of the 67 teleost and elasmobranch taxa, 16 (24%) were most abundant in the shallow stations and 50 (76%) were most abundant in the mid and deep stations. One species, Gray smoothhound Mustelus californicus, was not found in the nearshore stations.” (New Line 269) and “Of the 67 teleost and elasmobranch taxa, 46 (69%) had depth distribution patterns that did not vary across the three days (p>0.05; see S7 Table for p-values). Interestingly, 37 of these taxa (80%) are characterized as nonmigratory and non-transitory species (S7 Table). Of the 21 taxa with variable depth distributions (p<0.05), 10 (48%) of the taxa are transitory or migratory (S7 Table).” (New Lines 310-314)

In the Discussion, we added the sentence “Fifty-one (77%) of the taxa had eDNA depth distribution patterns that match their known depth preferences (pelagic vs. demersal).” (New Line 353). We removed the paragraph discussing the highest occurring taxa from the Discussion (Old Line 405) and added the sentence “Controlled field and lab experiments are needed to explicitly test this hypothesis.” (New Line 395). We also added a discussion of all species in the “eDNA signatures vary across time” section: “Of the 67 teleost and elasmobranch species, 47 (70%) had temporal variation in eDNA depth distributions that matched their known behavior…indicating eDNA may reflect behavioral patterns of marine vertebrates within the ecosystems they inhabit (S7 Table).” (New Lines 411-425). 

Line 174. I think it would be more accurate to say differences between kelp forest and surf zone, not "differences across horizontal space".

Thank you for this comment. We changed the sentence to “To analyze differences between the kelp forest and surf zone […]” (New Line 192).

line 189. It would help to better understand how ID works. There were 980 ASVs, and a total of 71 vertebrate species. What happened to all the ASVs? Is there a lumping threshold? What about likely contaminants such as human DNA? In Supplemental Methods S2 it refers to IDs with "55-59% confidence scores" as being "high confidence". That doesn't sound like a good match to me, percent identity might be more informative?

We added sentences to clarify the species ID process. We added the sentences “Second, we used the CRUX-generated 12S reference database supplemented with California Current Large Marine Ecosystem fish specific references to assign taxonomy using all available 12S reference barcodes to identify any non-fish taxa following the methods of Gold et al. [36] using a Bayesian cutoff score of 60. Although CRUX relies on ecoPCR (version 1.0.1) [37], blastn (version 2.6.0) [38], and Entrez-qiime (version 2.0) [28] as dependencies, we note that Bayesian cutoff scores are not directly analogous to percent identity from blastn. The BLCA classifier incorporates alignment metrics, including percent identity and percent overlap, into the underlying Bayesian model which then returns the Bayesian cutoff score metric as a measure of confidence for each taxonomic rank for a given ASV (See [36] and [28] for more detail). (New Lines 144-152)

We also added the sentences “We also manually removed sequences for species with taxonomic assignments for non-marine taxa (e.g. terrestrial mammals) in R. We then merged ASVs by summing reads by assigned taxonomy (e.g. summed all sequences reads from the 7 ASVs that assigned to Garibaldi, Hypsypops rubicundus).” (New Lines 158-161)

Line 202. This is a repeat of sentence on line 192.

We removed this sentence.

Line 211. I can guess but what do the box plot lines and whiskers represent?

We changed the caption to “Boxplots of mean species richness (� standard error) across depth stations” (New Line 228). 

Line 211. I think you should include that the surf zone data is not included in figure. It's slightly misleading since the surf zone data was used to calculate the indices.

Thank you for this suggestion. We added a sentence in the caption “Figure includes only the nearshore samples and does not include the surf zone samples.” (New Line 229)

Line 215. Can't you summarize p-values somehow in text, instead of referring reader to Supplementary Material?

We agree with this comment. We summarized p-values in main text: “Species richness differed significantly across depth, with shallow sampling stations having lower species richness than deeper stations (ANOVA; p<0.001; Fig 1). Specifically, the 0m and 1m species richness values were significantly different from the 9m (both p=0.001) and 10m (p=0.001 and p=0.002, respectively) species richness values. (New Line 222-227)

Also please be specific if sequences were matched to a local fish database, or to whatever is in GenBank.

We matched the sequences first using the California Current Large Marine Ecosystem fish specific reference database and then CRUX-generated 12S database. We describe the matching process in the Methods: “We first assigned taxonomy using the California Current Large Marine Ecosystem fish specific reference database [36]. Second, we used the CRUX-generated 12S reference database supplemented with California Current Large Marine Ecosystem fish specific references to assign taxonomy using all available 12S reference barcodes to identify any non-fish taxa following the methods of Gold et al. [36] using a Bayesian cutoff score of 60.” (New Lines 142-147)

Appendix S1 line 10. I don't understand how triplicate amplification helps "to account for PCR artifacts".

We removed “To account for PCR artifacts” from the sentence (S1 Appendix New Line 10)

---

## [Editor Report · Decision Letter 1]

13 Oct 2021

eDNA Captures Depth Partitioning in a Kelp Forest Ecosystem

PONE-D-21-17590R1

Dear Dr. Monuki,

We’re pleased to inform you that your manuscript has been judged scientifically suitable for publication and will be formally accepted for publication once it meets all outstanding technical requirements.

Kind regards,

Hideyuki Doi

Academic Editor

PLOS ONE

Additional Editor Comments (optional):

I carefully checked the revised manuscript as well as the response letter. I agree the revisions according to the reviewers’ comments and now can recommend to publish the paper in this journal.
---

## [Editor Report · Acceptance letter]

27 Oct 2021

PONE-D-21-17590R1 

eDNA Captures Depth Partitioning in a Kelp Forest Ecosystem 

Dear Dr. Monuki:

I'm pleased to inform you that your manuscript has been deemed suitable for publication in PLOS ONE. Congratulations! Your manuscript is now with our production department. 

Kind regards, 

on behalf of

Dr. Hideyuki Doi 

Academic Editor

PLOS ONE